# Suitability of Bronchoscopic Biopsy Tissue Samples for Next-Generation Sequencing

**DOI:** 10.3390/diagnostics11030391

**Published:** 2021-02-25

**Authors:** Shuji Murakami, Tomoyuki Yokose, Daiji Nemoto, Masaki Suzuki, Ryou Usui, Yoshiro Nakahara, Tetsuro Kondo, Terufumi Kato, Haruhiro Saito

**Affiliations:** 1Kanagawa Cancer Center, Department of Thoracic Oncology, Kanagawa 241-8515, Japan; ryousui@kcch.jp (R.U.); md100062@kcch.jp (Y.N.); tetsuro.kcc@gmail.com (T.K.); katote@kcch.jp (T.K.); saito-h@kcch.jp (H.S.); 2Kanagawa Cancer Center, Department of Pathology, Kanagawa 241-8515, Japan; yokose-t@kcch.jp (T.Y.); m-suzuki@kcch.jp (M.S.); 3Kanagawa Cancer Center, Department of Thoracic Surgery, Kanagawa 241-8515, Japan; nemotoda13@kcch.jp

**Keywords:** non-small-cell lung cancer, next-generation sequencing, bronchoscopy, Oncomine Dx Target Test

## Abstract

A sufficiently large tissue sample is required to perform next-generation sequencing (NGS) with a high success rate, but the majority of patients with advanced non-small-cell lung cancer (NSCLC) are diagnosed with small biopsy specimens. Biopsy samples were collected from 184 patients with bronchoscopically diagnosed NSCLC. The tissue surface area, tumor cell count, and tumor content rate of each biopsy sample were evaluated. The impact of the cut-off criteria for the tissue surface area (≥1 mm^2^) and tumor content rate (≥30%) on the success rate of the Oncomine Dx Target Test (ODxTT) was evaluated. The mean tissue surface area of the transbronchial biopsies was 1.23 ± 0.85 mm^2^ when small endobronchial ultrasonography with a guide sheath (EBUS-GS) was used, 2.16 ± 1.49 mm^2^ with large EBUS-GS, and 1.81 ± 0.75 mm^2^ with endobronchial biopsy (EBB). The proportion of samples with a tissue surface area of ≥1 mm^2^ was 48.8% for small EBUS-GS, 79.2% for large EBUS-GS, and 78.6% for EBB. Sixty-nine patients underwent ODxTT. The success rate of DNA sequencing was 84.1% and that of RNA sequencing was 92.7% over all patients. The success rate of DNA (RNA) sequencing was 57.1% (71.4%) for small EBUS-GS (*n* = 14), 93.4% (96.9%) for large EBUS-GS (*n* = 32), 62.5% (100%) for EBB (*n* = 8), and 100% (100%) for endobronchial ultrasound-guided transbronchial needle aspiration (EBUS-TBNA) (*n* = 15). Regardless of the device used, a tissue surface area of ≥ 1 mm^2^ is adequate for samples to be tested with NGS.

## 1. Introduction

Non-small-cell lung cancer (NSCLC) accounts for approximately 85% of lung cancer cases and is the leading cause of cancer-related deaths worldwide [1]. A large majority of patients with NSCLC are diagnosed at an advanced incurable stage, with a poor prognosis. Conventional cytotoxic chemotherapy has provided clinical benefit for or prolonged the survival of patients with advanced NSCLC, but the evidence for its clinical benefit is limited, with an objective response rate of 20–40% and a median overall survival (OS) of 10–14 months [2,3].

Targeted therapies have been developed to block aberrant oncogenic signaling and have dramatically changed the treatment strategies for advanced NSCLC patients with oncogenic driver mutations [4]. Targeted agents against oncogenic driver mutations such as sensitizing mutations in the epidermal growth factor receptor gene (*EGFR*) and fusions of echinoderm microtubule-associated protein-like 4 (*EML4*) and anaplastic lymphoma kinase (*ALK*), ROS proto-oncogene 1 (*ROS*1), or *BRAF* V600E have significantly improved the prognosis of advanced NSCLC, with higher response rates and longer progression-free survival than conventional cytotoxic chemotherapy [5,6,7,8,9,10]. Therefore, molecular biomarker testing has become crucial for treatment decision-making, especially for advanced NSCLC, to predict the efficacy of targeted therapies. A companion diagnostic technique has been included as part of the development of targeted therapies. Therefore, each companion diagnostic strategy is approved contemporaneously to select cancer patients for treatment with the approved novel targeted therapy. In daily clinical practice in Japan, each companion molecular test is simultaneously or sequentially performed for each targeted therapy: the cobas® EGFR Mutation Test v2 (Roche Molecular Systems, Pleasanton, CA, USA) for therapy with EGFR-tyrosine kinase inhibitors; the Vysis ALK Break Apart FISH Probe Kit (Abbott Molecular, Abbot Park, IL, USA) and the Ventana ALK (D5F3) CDx Assay (Roche Diagnostics, IN, USA) for treatment with ALK inhibitors; and the OncoGuide® AmoyDx ROS1 Gene Fusions Detection Kit (AmonyDiagnostics Co., Ltd., Xiamen, China) for treatment with ROS1 inhibitors. As the field of targeted therapies continues to evolve, the conventional paradigm in which a single drug is developed with a single companion diagnostic test is inevitably changing.

Next-generation sequencing (NGS) is a recently developed large-scale sequencing technology that has become a key technique for simultaneously screening multiple cancer-related genes [11]. The development of targeted therapies using NGS as a companion diagnostic test is also increasing. The Oncomine Dx Target Test (ODxTT) (Thermo Fisher Scientific, San Jose, CA, USA), which simultaneously evaluates 46 cancer-related genes, is one of the first NGS panels for NSCLC testing and was approved by the US Food and Drug Administration in June 2017 [12]. With this technology, the conventional paradigm in which a single drug is developed for a single companion diagnostic test to measure the variants of a single gene will soon evolve so that multiple drugs developed for clinical use are accompanied by a single NGS test as the companion diagnostic test.

Bronchoscopic biopsy is the preferred initial diagnostic procedure for suspected lung cancer [13]. The majority of patients with advanced NSCLC are diagnosed based on small biopsy specimens, which may be the only sample available in these patients. However, the limited number of tumor cells in small biopsy specimens can make molecular testing difficult. Sequential single-gene testing requires a large number of tissue slides, and determining the all-mutation status of targeted biomarkers will become increasingly difficult because these markers are expected to increase in the future [14]. NGS is a promising technology that can handle the increasing number of gene mutations that must be assessed. NGS requires fewer tissue slides than sequential single-gene testing, but the quality and quantity of nucleic acids in specimens must meet specific standards. In general, it is recommended that for ODxTT, the estimated tumor content ratio in a biopsy sample is > 30% of the total cells. After the tumor rate is satisfied, 10 or more slides of each tissue specimen must be examined. However, in practice, a sufficient amount of tissue is required to allow successful analysis with NGS, including ODxTT. We previously reported a correlation between the tissue surface area of a biopsy specimen and the success rate of ODxTT in detecting the *BRAF* V600E mutation and showed that the optimal cut-off value for this tissue surface area is 1.04 mm^2^ [15].

Transbronchial biopsy (TBB) with endobronchial ultrasonography with a guide sheath (EBUS-GS) is generally recognized as a useful method for the diagnosis of suspected lung cancer lesions. In clinical practice, two differently sized EBUS-GS kits are available, large EBUS-GS and small EBUS-GS. The number of tumor cells obtained with large EBUS-GS is considered to be greater than that obtained with small EBUS-GS [16]. In this study, we evaluated the size of the tumor sample collected with different bronchoscopic devices and the validity of the criteria used to select tumor samples for successful ODxTT.

## 2. Patients and Methods

### 2.1. Patients and Study Design

The study participants included 184 patients with NSCLC who were pathologically diagnosed with a biopsy sample between September 2019 and May 2020 at the Kanagawa Cancer Center Hospital, Yokohama, Japan. The biopsy procedures used included TBB with EBUS-GS, endobronchial biopsy (EBB) under direct-vision forceps, and endobronchial ultrasound-guided transbronchial needle aspiration (EBUS-TBNA). We retrospectively reviewed the medical records of all the patients included in this study and analyzed the pathological features of their biopsy samples. We obtained ethical approval from the Kanagawa Cancer Center Hospital, Japan (2019EKI-48), and patient confidentiality was maintained. The TBB samples were obtained with small (small EBUS-GS; FB-233D; Olympus Medical Systems, Tokyo, Japan) or large forceps (large EBUS-GS; FB-231D; Olympus Medical Systems). The EBB samples were obtained with biopsy forceps (Radial Jaw 4; Boston Scientific Corporation, Natick, MA, USA). The EBUS-TBNA samples were obtained with 22-gauge (22G) aspiration needles (Expect™ Pulmonary E00558220, Boston Scientific Corporation).

### 2.2. Pathological Diagnosis and Evaluation of Pathological Factors

The biopsy samples were fixed in 10% neutral-buffered formalin solution for 6–24 h, embedded in paraffin wax, and processed for histopathological examination with routine histological techniques. The histological diagnoses were made by two pathologists (M.S. and T.Y.) according to the 2015 World Health Organization (WHO) Classification of Tumors of the Lung [17].

We assessed three pathological factors that potentially influence the success of NGS analysis: tissue surface area, tumor cell count, and tumor content ratio. The tissue surface area (mm^2^) was defined as the area occupied by the tumor tissue on a slide and was measured with the NIS-Elements D version 4.60 image analysis software (Nikon, Japan). The tumor cell count (cells) was defined as the number of tumor cells on a slide and was estimated by multiplying the tissue surface area by the number of tumor cells per 1 mm^2^, which was counted with an ocular micrometer. The tumor content ratio was evaluated as the number of tumor cells among all the nucleated cells on a slide and was scored in approximately 5% increments by the pathologists.

### 2.3. Biomarker Analysis

After the morphological diagnosis, therapy-predictive biomarker testing for NSCLC was routinely performed when there was sufficient tissue. Testing for *EGFR* and *BRAF* mutations and rearrangements involving the *ALK* and *ROS1* genes is now considered routine clinical practice. Testing for these four driver mutations involves individual stand-alone tests or a multiplex PCR-based NGS assay. In general, it is recommended that for ODxTT, the estimated tumor content ratio in a biopsy sample is greater than 30% of the total cells. Since January 2020, we have added a tumor area of ≥ 1 mm^2^ to the recommended local criteria. The ODxTT, a multiplex PCR-based NGS assay, was performed for the clinical companion diagnosis of routine driver mutations. Ten 5 μm-thick sections were sectioned from each biopsy sample for testing with ODxTT.

### 2.4. ODxTT in Patients with NSCLC

ODxTT was used to sequence the tagged amplicons of hotspots and the coding regions of 46 genes. DNA and RNA were extracted from NSCLC tissue samples prepared as formalin-fixed paraffin-embedded sections on slides. Either DNA (10 ng) or RNA (10 ng) from the tissue samples was required for each target amplification reaction. Samples with <10 ng of extracted DNA or RNA were excluded from the analysis as “insufficient quantity.” If the extracted DNA or RNA did not meet the sequencing quality control (QC) metrics, the samples were excluded from the analysis as “invalid.” The RNA was enzymatically converted to complementary DNA (cDNA) and the target regions of interest in the cDNA or DNA were specifically amplified with PCR, and libraries were generated. After template preparation, sequencing by synthesis was performed to determine the nucleic acid sequences of the amplified libraries. The DNA and RNA reads were mapped to the human reference genome to detect the single-nucleotide variants or deletions in 46 genes in the DNA and fusions in 21 RNAs. If the number of reads was insufficient or the background noise was large, the result was not considered appropriate and it was excluded as “no call.”

The effective analysis of the DNA sequences of *EGFR* and *BRAF* V600E was deemed a “success.” The effective analysis of the *ALK* and *ROS1* RNA sequences was deemed a “success.” If the result for one of the four driver mutations was “invalid” or “no call,” it was deemed to be unsuccessful.

### 2.5. Statistical Analysis

The tissue surface area, tumor cell count, and tumor content ratio in the subsets of patients were expressed as mean values ± standard deviations (SD) and were compared across subgroups with the Student’s *t*-test. The success rates of ODxTT in the subsets of patients were compared with the χ^2^ test. Statistical significance was defined as a *p*-value of less than 0.05. Statistical analyses were performed with SPSS v24 software (SPSS Inc., Chicago, IL, USA).

## 3. Results

### 3.1. Patient Characteristics

A total of 184 patients who were diagnosed bronchoscopically with NSCLC were reviewed in this study. The patient characteristics are summarized in Table 1. The mean age of the patients was 72 years. Of the 184 patients, 134 were men, and 133 were histologically diagnosed with non-squamous cell carcinoma. The distribution by stage was as follows: stage I, 23 patients; stage II, 17 patients; stage III, 54 patients; stage IV, 84 patients; and postoperative recurrence, 6 patients. Tumor tissues were sampled with large EBUS-GS (*n* = 43), small EBUS-GS (*n* = 77), EBB (*n* = 14), or EBUS-TBNA (n = 50), and 165 patients were initially diagnosed with bronchoscopy. 

### 3.2. Tumor Content Determined in Biopsy Samples

The mean tumor content ratios were 32.2 ± 19.6% in the small EBUS-GS subgroup, 31.7 ± 17.4% in the large EBUS-GS subgroup, 46.4 ± 28.1% in the EBB subgroup, and 26.2 ± 17.3% in the EBUS-TBNA subgroup (Figure 1a). The tumor content ratio was higher in the EBB subgroup (*p* = 0.010) and lower in the EBUS-TBNA subgroup (*p* = 0.084) than in the large EBUS-GS subgroup, but did not differ significantly between the small and large EBUS-GS subgroups (*p* = 0.881). The proportion of patients with a tumor content ratio > 30% was 55.8% in the small EBUS-GS subgroup, 51.9% in the large EBUS-GS subgroup, 64.3% in the EBB subgroup, and 36.0% in the EBUS-TBNA subgroup (Table 2).

The mean tumor cell counts were 316 ± 242 cells in the small EBUS-GS subgroup, 394 ± 280 cells in the large EBUS-GS subgroup, 482 ± 482 cells in the EBB subgroup, and 809 ± 867 cells in the EBUS-TBNA subgroup (Figure 1b). The tumor cell count was slightly higher in the large EBUS-GS subgroup than in the small EBUS-GS subgroup, but the difference was not significant (*p* = 0.128). It was also significantly higher in the EBUS-TBNA subgroup than in the small EBUS-GS and large EBUS-GS subgroups (*p* < 0.001 and *p* = 0.001, respectively).

The mean tissue surface area was 1.23 ± 0.85 mm^2^ in the small EBUS-GS subgroup, 2.16 ± 1.49 mm^2^ in the large EBUS-GS subgroup, and 1.81 ± 0.75 mm^2^ in the EBB subgroup (Figure 1c). The tissue surface area was larger in the EBB subgroup (*p* = 0.027) and the large EBUS-GS subgroup (*p* < 0.001) than in the small EBUS-GS subgroup. The proportion of patients with a tissue surface area > 1 mm^2^ was 48.8% in the small EBUS-GS subgroup, 79.2% in the large EBUS-GS subgroup, and 78.6% in the EBB subgroup (Table 2). The proportion of patients with a tumor content ratio > 30% and a tissue surface area > 1 mm^2^ was 41.7% in the small EBUS-GS subgroup, 42.9% in the large EBUS-GS subgroup, and 42.9% in the EBB subgroup (Table 2).

### 3.3. Success Rate of ODxTT Using Different Bronchoscopic Devices

Sixty-nine of the 184 patients who were pathologically diagnosed with NSCLC were tested with ODxTT (37.5%) (Figure 2). The success rate of DNA sequencing was 84.1% (58/69), that of RNA sequencing was 92.7% (64/69), and that of both types of sequencing was 82.6% (57/69) across all patients. The success rate of DNA sequencing was 57.1% (8/14) in the small EBUS-GS subgroup, 93.4% (30/32) in the large EBUS-GS subgroup, 62.5% (5/8) in the EBB subgroup, and 100% (15/15) in the EBUS-TBNA subgroup. The success rate of RNA sequencing was 71.4% (10/14) in the small EBUS-GS subgroup, 96.9% (31/32) in the large EBUS-GS subgroup, 100% (8/8) in the EBB subgroup, and 100% (15/15) in the EBUS-TBNA subgroup.

### 3.4. Success Rate of ODxTT by Criteria

The success rate of DNA sequencing was 82.5% (52/63) among tumors with a tumor content ratio ≥ 30%; 91.8% (45/49) among tumors with a tumor content ratio ≥ 30% and a tissue surface area ≥ 1 mm^2^; and 58.8% (10/17) among tumors without a tumor content ratio ≥ 30% and a tissue surface area ≥ 1 mm^2^ (Figure 3). The success rate of RNA sequencing was 92.1% (58/63) among tumors with a tumor content ratio ≥ 30%; 98.0% (48/49) among tumors with a tumor content ratio ≥ 30% and a tissue surface area ≥ 1 mm^2^; and 76.5% (13/17) among tumors without a tumor content ratio ≥ 30%, and a tissue surface area ≥ 1 mm^2^. Moreover, the success rate was 100% with both DNA and RNA sequencing for the tumors with a tumor content ratio ≥ 30% and a tissue surface area ≥ 1 mm^2^ in the large EBUS-GS subgroup.

## 4. Discussion

NGS testing will become increasingly important for detecting driver mutations in advanced NSCLC. A tumor content rate of > 30% has been recommended for the use of ODxTT. However, in the present study, a tumor content rate of > 30% was observed in only about 50% of NSCLC tumor samples obtained with bronchoscopy. The size of a tumor sample obtained with this method depends on the device; tumor samples obtained with large EBUS-GS are about twice as large as those obtained with small EBUS-GS. The success rate of DNA sequencing was 84.1% and that of RNA sequencing was 92.7% across all patients. The success rate of ODxTT was higher in the samples obtained with large EBUS-GS (94% for DNA and 97% for RNA) and EBUS-TBNA (100% for both DNA and RNA). The failure rate of ODxTT was reduced by adding a tumor sample size of > 1 mm^2^ as a criterion for sample selection. All of the samples obtained with EBUS-GS that met the criterion were successfully analyzed with ODxTT.

Upfront genotyping at the time of diagnosis is an essential step in selecting the optimal first-line therapy for NSCLC. The use of matched targeted treatments in individuals with oncogenic driver mutations improves the survival of those with several types of driver mutations [18]. Cytotoxic chemotherapy is no longer the standard first-line treatment for patients with driver mutations, except for those with backgrounds for which targeted therapy should be avoided. In addition to therapies that target EGFR, ALK, ROS1, and BRAF, inhibitors of MET have recently been approved in Japan. Furthermore, it will be necessary to detect rare mutations, such as in *KRAS* and *RET*, in the near future. However, in clinical practice, biomarker testing cannot be adequately performed in some patients with advanced NSCLC, mainly because the sample volume is too small. In the BRAVE study, a multicenter observational study of biomarker testing and first-line treatment for advanced NSCLC patients in Japan, the proportion of patients with a confirmed biomarker status, including for *EGFR*, *ALK*, *ROS1*, and *PD-L1*, for use in selecting the first-line treatment was 79.7% [19]. Another study showed that as the number of single-gene tests for each driver mutation increases, large numbers of tissue slides are required and the number of single tests ordered decreases [14]. To resolve these clinical problems, the widespread use of upfront NGS to test all known lung cancer-related genes at the time of diagnosis is essential. However, a sufficient size of a tissue sample is required to perform NGS with a high success rate. In the present study, only 37.5% of NSCLC patients who were diagnosed with bronchoscopy could be evaluated with NGS.

Several studies have focused on the feasibility of using bronchoscopy-collected tissues for NGS analyses. The success rate of NGS using TBB is reported to be relatively low, at 15–82% [20,21]. It should be noted that the number of genes in the NGS panels used differed in each report. ODxTT can simultaneously evaluate changes in 46 driver genes, but the number of genes detected is smaller than the number detected with a comprehensive genome profile test. Therefore, NGS analyses that can detect large numbers of genes require large tumor samples, and the success rate of NGS using small samples can be low [20]. In a previous study using ODxTT, all 18 samples collected with TBB and cryobiopsy were considered successful [22]. The mean total area of the samples was 8.5 mm^2^ (3.6–32.0 mm^2^). Cryobiopsy has been reported to provide larger tissue specimens than forceps biopsy, resulting in a high NGS success rate [20]. Cryobiopsy was not used in the present study, so the tumor surface areas were smaller than those taken with cryobiopsy in previous studies. In the present study, the mean tumor surface area in the large EBUS-GS samples was significantly greater than that in the small EBUS-GS samples and the success rate of NGS in the large EBUS-GS group was correspondingly higher than that in the small EBUS-GS group. Similar to previous reports, our results suggest that taking larger samples increases the success rate of NGS. Our results indicate that larger tissue samples should be collected to ensure successful NGS analyses, although the minimum tumor sample size required for ODxTT is still unclear.

In a previous work, we examined the correlation between the tumor surface area and the success rate of ODxTT, and reported that a tumor surface area ≥ 1 mm^2^ was the best discriminative value to identify samples for ODxTT analyses [15]. In the present study, the success rate of NGS in samples with a tumor surface area ≥ 1 mm^2^ was higher than that in samples with a tumor surface area < 1 mm^2^. Moreover, the fact that the success rate was 100% for both DNA and RNA sequencing for samples with a tissue surface area ≥ 1 mm^2^ in the large EBUS-GS group suggests that the tumor surface area is a valid criterion for ODxTT analysis.

The tumor content ratio is another factor affecting sample suitability for the ODxTT. If the tumor content ratio is < 30%, it is recommended that the tissue sample be macrodissected and its tumor content enriched. However, macrodissection is usually difficult for small samples such as bronchoscopic samples. The fact that a tumor count ratio of ≥ 30% was observed in only about 50% of NSCLC tumor samples obtained with bronchoscopy is a major obstacle to expanding NGS analyses to bronchoscopic tissues. The tumor content ratio of the EBUS-TBNA samples was lower than that of the EBUS-GS and EBB samples, and the proportion of patients with a tumor content ratio of ≥ 30% was only 36.0% in the EBUS-TBNA subgroup. Blood contamination was detected in the EBUS-TBNA samples, which could have affected the tumor content ratio [23,24]. The low tumor content ratio in the EUBS-TBNA samples and the high concentration of crushed tumor cells could reduce the success rate of NGS analyses [22]. The success rate of the ODxTT on the EBUS-TBNA samples was high (100%) in the present study. The rate of selection for analysis with the ODxTT was only 30% in the present study. The samples analyzed with the ODxTT were selected by the pathologist based on the concentration of crushed tumor cells.

This study had several limitations. Firstly, it was based on the retrospective review of medical records. There was potential for selection bias because the study was conducted at a single institution and we only included patients for whom a pathological diagnosis was established with bronchoscopy alone. Therefore, patients who were diagnosed with cytology alone or for whom diagnosis failed were not included in the study. Secondly, the ODxTT was predominantly performed on tumor samples that met the criteria of our institution. Therefore, the success rate of NGS in the tumor samples that did not meet the criteria was inaccurate. As long as we analyzed the samples with tumor content rates of ≥ 30% in the present study, we could not adequately determine the relationship between the tumor count rate and the success rate of NGS. A cut-off value of 30% may severely constrain the rate of samples selected for NGS analysis. Indeed, in the present study, a tumor count rate of > 30% was only observed in about 50% of patients. Finally, the tissue surface area of tumor samples collected with EBUS-TBNA was not measured, so the selection of these samples for NGS analysis was not dependent on this criterion. Only selected samples were submitted for NGS analysis, so the success rate of NGS for tumor samples collected with EBUS-TBNA was inaccurate.

Despite these limitations, the finding that an NGS analysis can be successful with a relatively small sample of ≥ 1 mm^2^ may be useful in clinical practice, and the criterion based on the tissue surface area can guide the selection of tumor samples for NGS analysis. Increasing the NGS analysis rate is a future goal, so the size of tumor samples required in cases where the tumor count rate is < 30% must be established.

## 5. Conclusions

A sufficient size of a tumor sample is required for successful NGS testing. The size of a specimen that can be obtained with bronchoscopy is limited, but samples with a tissue surface area of ≥1 mm^2^ may be adequate for NGS testing. To provide matched targeted treatments based on NGS analyses, clinicians must collect the largest samples possible using large EBUS-GS.

## Figures and Tables

**Figure 1 diagnostics-11-00391-f001:**
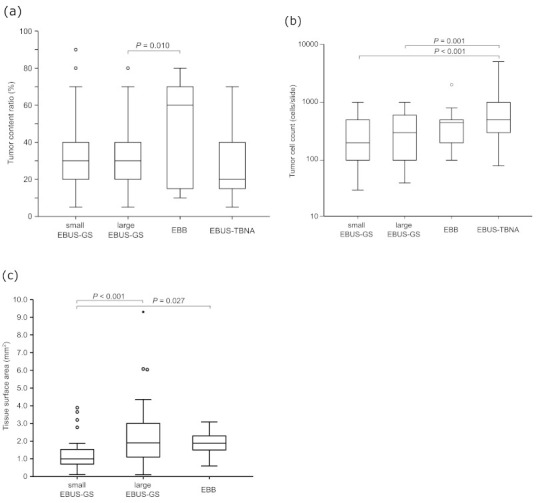
(**a**) Tumor content ratio (%), (**b**) tumor cell count (cells), and (**c**) tissue surface area (mm^2^) for each bronchoscopic device. EBUS-GS, endobronchial ultrasonography with a guide sheath; EBB, endobronchial biopsy; EBUS-TBNA, endobronchial ultrasound-guided transbronchial needle aspiration.

**Figure 2 diagnostics-11-00391-f002:**
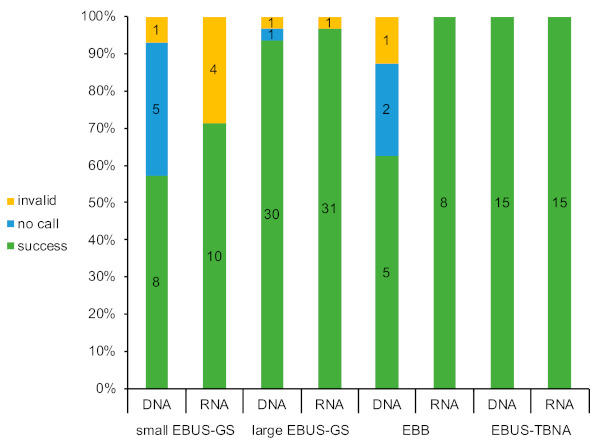
Success rate of the Oncomine Dx Target Test (ODxTT) for each bronchoscopic device. EBUS-GS, endobronchial ultrasonography with a guide sheath; EBB, endobronchial biopsy; EBUS-TBNA, endobronchial ultrasound-guided transbronchial needle aspiration.

**Figure 3 diagnostics-11-00391-f003:**
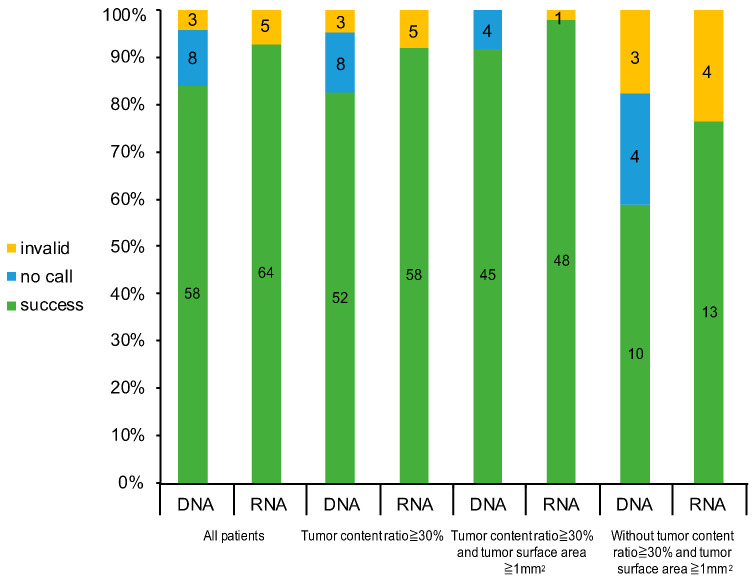
Success rate of the Oncomine Dx Target Test (ODxTT) in the specimens that met the tumor sample criteria.

**Table 1 diagnostics-11-00391-t001:** Patient characteristics.

Characteristic	All Patients (*n* = 184)	Analyzed with ODxTT (*n* = 69)
Median age, years (range)	72 (42–88)	72 (45–88)
Sex (male/female)	134/50	55/14
Tumor stage (I/II/III/IV/r)	23/17/54/84/6	6/9/23/30/1
Histology (SQ/non-SQ)	51/133	27/42
Bronchoscopic device		
Small EBUS-GS	43	14 (32.6%)
Large EBUS-GS	77	32 (41.5%)
EBB	14	8 (57.1%)
EBUS-TBNA	50	15 (30%)
Purpose of biopsy (initial/second biopsy)	165/19	66/3

ODxTT, Oncomine Dx Target Test; r, recurrence; SQ, squamous cell carcinoma; EBUS-GS, endobronchial ultrasonography with a guide sheath; EBB, endobronchial biopsy; EBUS-TBNA, endobronchial ultrasound-guided transbronchial needle aspiration.

**Table 2 diagnostics-11-00391-t002:** Number (proportions) of specimens that met the tumor amount criteria for each bronchoscopic device.

Bronchoscopic Device	Small EBUS-GS	Large EBUS-GS	EBB	EBUS-TBNA
n	43	77	14	50
Tumor content ratio ≥ 30%	24 (55.8%)	40 (51.9%)	9 (64.3%)	18 (36%)
Tissue surface area ≥ 1 mm^2^	21 (48.4%)	61 (79.2%)	11 (78.6%)	-
Tumor content ratio ≥ 30% and tissue surface area ≥ 1 mm^2^	10 (41.7%)	33 (42.9%)	6 (42.9%)	-

EBUS-GS, endobronchial ultrasonography with a guide sheath; EBB, endobronchial biopsy; EBUS-TBNA, endobronchial ultrasound-guided transbronchial needle aspiration.

## Data Availability

Data are available from the corresponding author upon reasonable request.

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
