# Peer review of "Suitability of Bronchoscopic Biopsy Tissue Samples for Next-Generation Sequencing"

_diagnostics, 2021, doi:10.3390/diagnostics11030391_

Round 1

Reviewer 1 Report

The manuscript presented by Marukami et al. is a follow-up of a previous work (doi:10.1111/1759-7714.12743) where they described an optimal tissue surface area (biopsy) cut-off value (>= 1mm2) for mutation sequencing. This new criteria is now implemented since January 2020. Now the authors investigated whether standard recommendations for OdxTT and this new parameter influence NGS analysis in NSCLC samples obtained from different biopsy procedures. The outocomes of this study indicate combination of both criteria can improve the success rates of NGS, although in a small number of patients. The authors are aware of the limitations of this study and are well present in the manuscript. Overall, the manuscript is clear, informative and well written.

Minor point:

1- Figure 1: is missing the lettering (a, b, c).

2- Figure 1 and 2: Please provide only the statistical significance (p values < 0.05) in the figure. It will be easier to the readers to understand which parameters are being compared. 

3 – In subsection “3.4 Success rate of OdxTT by criteria” the numbers did not match with Figure 3. For instance the authors described that: the success rate of DNA sequencing was “89.2% (33/37) among tumors with a tumor content ratio >=30% and a tissue surface are >= 1mm2” and of 93.3% (36/37) for RNA. However, in Figure 3 they show that 45/49 samples were successfully sequenced for DNA and 48/49 for RNA. Please clarify this and check the numbers and percentages.

4- The authors also mentioned that 10/17 (DNA) were successfully sequenced in the subgroup: tumor content >=0 and area >=1mm2 (figure 3). The same statement is present for RNA sequencing analysis. This information is not present in Figure3. Please clarify.

Indeed, the work will benefit if the authors make a comparison of sequencing data between: >30% cancer cells vs >30% + >1mm2 vs >1mm2 only).

5 – The increased success rate in combining both criteria is due to the exclusion of samples that already did not worked in >=30% criteria alone? If yes, adding >=1mm2 may not be a critical factor for an increased success rate but, instead, may be due to the elimination of cases that independently of the criteria will not work at all (problem with sequencing/DNA,RNA quality). Could the authors comment on that?

6- It is missing some information in some references (as the case of ref 15 where pages are missing).

Author Response

Response to the comment from Reviewer #1

Minor point:

  • Figure 1: is missing the lettering (a, b, c).

Answer: In accordance with the Reviewer’s comment, we added the lettering (a, b, c) in the figure1

  • Figure 1 and 2: Please provide only the statistical significance (p values < 0.05) in the figure. It will be easier to the readers to understand which parameters are being compared. 

Answer: In accordance with the Reviewer’s comment, we added p values in the figure1.

3 – In subsection “3.4 Success rate of OdxTT by criteria” the numbers did not match with Figure 3. For instance the authors described that: the success rate of DNA sequencing was “89.2% (33/37) among tumors with a tumor content ratio >=30% and a tissue surface are >= 1mm2” and of 93.3% (36/37) for RNA. However, in Figure 3 they show that 45/49 samples were successfully sequenced for DNA and 48/49 for RNA. Please clarify this and check the numbers and percentages.

Answer: The reviewer's comment is correct. I re-checked and corrected the numbers and percentages.

4- The authors also mentioned that 10/17 (DNA) were successfully sequenced in the subgroup: tumor content >=0 and area >=1mm2 (figure 3). The same statement is present for RNA sequencing analysis. This information is not present in Figure3. Please clarify. Indeed, the work will benefit if the authors make a comparison of sequencing data between: >30% cancer cells vs >30% + >1mm2 vs >1mm2 only).

Answer: Because I found a mistake, I corrected ">=0%" to ">=30%". In accordance with the Reviewer’s comment, we added the data of DNA and RNA sequencing analysis of the subgroup without tumor content>=30 and area >=1mm2 in the figure3.

5 – The increased success rate in combining both criteria is due to the exclusion of samples that already did not worked in >=30% criteria alone? If yes, adding >=1mm2 may not be a critical factor for an increased success rate but, instead, may be due to the elimination of cases that independently of the criteria will not work at all (problem with sequencing/DNA,RNA quality). Could the authors comment on that?

Answer: As the reviewer’s comment, the quantity and quality of DNA/RNAN is important to the success of sequencing, but we think that a sufficient amount of sample is more important. In this study we focus only on the problem of sample volume. We did not change the inspection method that would affect the quality during this research period, therefore we believe that an increased success rate was not due to the improvement of DNA/RNA quality. Of course, we think it will be necessary in the future to evaluate the quality of failed cases.

6- It is missing some information in some references (as the case of ref 15 where pages are missing).

Answer: The reviewer's comment is correct. We fixed References 15.

Reviewer 2 Report

The title introduces the main topic. The article is finalized to help clinicians to perform the correct experimental procedures to allow a clear molecular diagnosis and classifications of non-small-cell lung cancer (NSCLC). The correct molecular diagnosis allows performing the best therapy.

The abstract is simple and straightforward.

The introduction is focused. The authors clearly explain what is already known about this topic and why it is crucial to perform a correct diagnosis, based on identifying the mutated genes, to allow a correct therapy, thus to achieve the best prognosis.

The methods are clearly explained and the results are exhaustive.

Figures and tables well report the results.

Results are critically discussed in the discussion, and the study's limitations are explained and represent a good starting point to improve future research.

Minor revisions

Figure 1. Please, add significance.

Limitations of the study:

The sample size needs to be increased, not necessarily within this study, representing a starting point.

Overall, the article is clear and well written. 

Author Response

Response to comment from Reviewer 2

Minor revisions

Figure 1. Please, add significance.

Answer: In accordance with the Reviewer’s comment, we added p values in the figure1.
